# Opening up the Toolbox: Synthesis and Mechanisms of Phosphoramidates

**DOI:** 10.3390/molecules25163684

**Published:** 2020-08-13

**Authors:** Emeka J. Itumoh, Shailja Data, Erin M. Leitao

**Affiliations:** 1School of Chemical Sciences, The University of Auckland, 23 Symonds Street, Auckland 1010, New Zealand; eitu709@aucklanduni.ac.nz (E.J.I.); sdat751@aucklanduni.ac.nz (S.D.); 2Department of Industrial Chemistry, Ebonyi State University, Abakaliki 480001, Ebonyi State, Nigeria; 3The MacDiarmid Institute for Advanced Materials and Nanotechnology, Wellington 6140, New Zealand

**Keywords:** phosphoramidate, synthetic routes, mechanism, applications

## Abstract

This review covers the main synthetic routes to and the corresponding mechanisms of phosphoramidate formation. The synthetic routes can be separated into six categories: salt elimination, oxidative cross-coupling, azide, reduction, hydrophosphinylation, and phosphoramidate-aldehyde-dienophile (PAD). Examples of some important compounds synthesized through these routes are provided. As an important class of organophosphorus compounds, the applications of phosphoramidate compounds, are also briefly introduced.

## 1. Introduction to Phosphoramidates and their Applications

Phosphoramidates (P-N) are a class of organophosphorus compounds known for the presence of a single covalent bond between the tetracoordinate P(V) atom and N(III) atom. There are generally three types of phosphoroamidates, which are distinguished according to the substitution on the P and N atoms (Figure 1) [1].

Another feature of these organophosphorous compounds, defined as (RO)_2_P(O)NR’_2_ (R, R’ = H, alkyl, aryl, heteroaryl), is a stable phosphoryl bond (P=O). Both P and N atoms are key physiological elements present in genetic material, energy transfer, enzymes, and other biomolecules and are required for various life processes. As such, molecules containing P-N linkages are found in a large array of biologically active natural products (**1**–**7**) (Figure 2) [2,3]. For example, Microcin C7 (**1**) (Figure 2) is an antibiotic produced by *Escherichia coli* [4]. Dinogunellin (**2**) (Figure 2), which is produced in the roe of some fishes, is a natural toxin [5]. Phosphoarginine and phosphocreatine (**3** and **6**) (Figure 2) are important biological molecules used as sources of stored energies in invertebrates and vertebrates, respectively [6]. Phosphoramidon (**4**) (Figure 2), derived from *Streptomyces tanashiensis*, inhibits the thermolysin enzyme, which is a key factor in the development of various diseases [7,8]. Lastly, phosmidosine and agrocin 84 (**5** and **7**) (Figure 2) are nucleotide antibiotics isolated from *Streptomyces durhameusis* and *Agrobacterium radiobacter* and are used to control gray mold disease and crown gall disease in plants, respectively [9,10].

Due to the inherent physiochemical properties of the phosphorus atom including its polarizability, multivalency, varying oxidation states, and low coordination number, a diverse range of compounds containing the P-N motifs have been synthesized [11]. These P-Ns are widely used in medicine for the control and treatment of various diseases, in agriculture as pesticides for crop protection, in industry as novel fire-retardant compounds to delay the flammability of polymeric compounds, and in the fields of analytical and coordination chemistry. Therefore, synthetic P-N compounds are not only researched out of academic interest, but they also have commercial applications. Uses for P-Ns can be separated into 5 main classes: agriculture, industry, analytical chemistry, synthetic chemistry, and medicinal chemistry (Figure 3) [12,13,14,15,16,17,18].

### 1.1. Phosphoramidates in Agriculture

This class of organophosphates (with common names such as cruformate, fenamiphos, and fosthietan) has been widely used in pest management for crop protection (Figure 3) [19,20,21]. P-Ns for pest control affect the nervous system of pests/insects by inhibiting acetylcholinesterase (AChE), a major neurotransmitter [22,23,24,25]. In addition, these P-N compounds have been utilized as urease inhibitors [26,27]. Urease is a nickel-based enzyme that catalyzes the hydrolysis of urea on soil surfaces into volatile ammonia and carbon dioxide (Scheme 1) [28,29]. Thus, on inhibiting the activity of this metalloenzyme, the availability and performance of urea is enhanced, which is a vital nitrogenous fertilizer for plant growth.

### 1.2. Phosphoramidates in Industry

P-Ns have been used as additives in both oil and water-based lubricants (Figure 3). It has been proposed that phosphorus acts as a key element in enhancing the tribological properties of water/oil-based fluids by forming a protective layer containing phosphates and/or polyphosphates. Thus, they possess anti-wear and friction-reducing properties [18,30]. Furthermore, these P-Ns have been found to be a promising alternative to halogenated flame retardants (FRs), which are being phased out as they have been recognized as global contaminants due to their ill effects on health and the environment [31]. P-N fire retardants are less volatile and relatively thermally stable with enhanced char formation, and they are more efficient and sustainable than their halogenated counterparts [32,33]. Recently, compounds containing both P and N atoms in combination have been reported to have a synergistic effect, as they can act in both vapor phase and condensed phase, thus offering improved flame retardancy on various substrates such as polyurethane foams, cotton, cellulose, epoxy resins, textiles, etc. [34,35,36,37,38,39,40]. For example, novel P-N-based intumescent flame retardants have been incorporated into polycarbonates (PC) to impart flame retardancy to the composite by forming an expandable protective char [41].

### 1.3. Phosphoramidates in Analytical Chemistry

MALDI-TOF MS is a prominent tool for analyzing biomolecules (Figure 3) [42]. For the detection of low molecular weight such as amino acids and peptides, conventional matrices yield a lot of matrix-related ions in the low *m/z* range of the spectrum, thus interfering with analyte signals. To tackle this, various approaches such as using additives in the organic matrix, high mass matrix molecules, or matrices based on inorganic compounds have been employed [43]. Furthermore, to improve analysis, synthesizing derivatives of small analyte molecules by *N*-phosphorylation labeling has been done. The neutral phosphoryl group attached at the *N*-terminal of amino acids suppresses the matrix signals and favors the transfer of energy from matrix to analyte for ionization by secondary ion-molecule reactions. Therefore, P-Ns have been used for analysis of biomolecules to improve the ionization efficiency and suppress matrix background signals [44]. In addition to this, P-Ns have found novel applications as task-specific ionic liquids (TSILs), which are tailored ionic liquids having a specific functionality to complex metal ions. Recently, these TSILs have been used for the efficient extraction of uranium metal, which is an important element in nuclear fuel, and the extraction of uranium metal from the spent/used nuclear fuel is vital for the sustainable economy [45]. The enhanced extraction capabilities using TSILs in contrast to liquid–liquid extraction could be attributed to synergistic effects as a result of complexation and H-bonding [17]. To this end, these tailored ionic liquids have been found to be selective for the micellar extraction of uranium metal from aqueous waste samples as even the ultra-trace levels of uranium are toxic [46].

### 1.4. Phosphoramidates in Synthetic Chemistry

P-Ns act as 1,3-*N*,*O*-chelating ligands (e.g., Figure 3, Ligand) and have diverse applications in the field of coordination chemistry for bond activation, catalysis, and metal ligand cooperativity [47,48,49]. For example, a P-N-tantalum complex has been developed as a pre-catalyst for hydroaminoalkylation reactions [50]. In addition, the synthesis of medium and large N-based heterocycles has been achieved with the aid of P-Ns, as the phosphoryl group improves intramolecular cyclization, and it can also be easily removed afterwards [51]. Furthermore, P-Ns have been used as directing groups for arylation reactions, as they selectively activate C-H bonds, and thus, desired functionality can be achieved [52].

### 1.5. Phosphoramidates in Pharmaceutical and Medicinal Chemistry

P-Ns have garnered considerable interest in the field of biomedicine due to their potential applications as antimicrobial [53,54,55], antioxidant [56], anticancer [57], antimalarial [58], antiviral [59,60], and anti-HIV [61] drugs and prodrugs against various ailments (Figure 3). The ProTide approach, pioneered by the McGuigan group in 1992 for drug delivery, is a powerful tool to enhance the efficacy, intracellular delivery, and therapeutic potential of the parent drugs [62,63]. Recently, remdesivir (Figure 3, drug candidate), which has a P-N moiety, has been evaluated as a therapeutic option for the treatment of COVID-19-affected patients and has attracted a lot of attention in the clinical field [64]. Additionally, these P-N compounds have been used in gene therapy and nucleoside treatment [65,66,67]. Moreover, many P-Ns have been found to be hydrolytically degradable under acidic conditions and are comparatively stable at neutral and higher pHs. The acid labile P-N bond is selectively cleaved due to protonation at N, which then undergoes nucleophilic attack by a molecule of water to eliminate the corresponding phosphates and amines [68,69]. Thus, these pH responsive P-Ns have notable applications in controlled drug release [70,71]. For example, the P-N-based prodrug of l-Dopa has been synthesized for its controlled release for the treatment of Parkinson’s disease [72]. Similarly, P-N derivatives of the antiherpetic drug, acyclovir (ACV), hydrolyses at pH 2 into ACV monophosphate and can be a promising antiviral drug due to increased cell penetration and therapeutic potential relative to the parent drug [73].

## 2. Synthetic Routes to Phosphoramidates

Although P-N biomolecules such as phosphocreatine have been known since the work of Fiske and Subbarow in 1927, synthetic routes to P-N compounds prior to 1988 have been limited [74,75]. These early routes involve the preparation of and use of toxic reagents over multiple synthetic steps and often resulted in the elimination of stoichiometric waste. However, with an increase in the number of uses of P-Ns, the need to develop one-pot synthetic routes that are more environmentally benign has driven researchers to discover catalytic strategies. Accordingly, all the known synthetic routes to P-Ns can be separated into six categories: salt elimination, oxidative cross-coupling, azide, reduction, hydrophosphinylation, and phosphoramidate-aldehyde-dienophile (PAD). Each of the categories will be discussed in detail in the coming sections, with an emphasis on conditions, scope, and mechanism. This discussion will be preceded by a brief introduction to the early synthetic routes.

### 2.1. A Brief Historical Timeline of Phosphoramidate Synthesis

Based on the available literature, one of the earliest syntheses of P-Ns involved the works of Audrieth and Toy [76,77]. Audrieth and Toy treated phosphoryl trichloride with phenol in the presence of pyridine to obtain a mixture of diphenylchlorophosphate (*d*PCP) and phenyl dichlorophosphate (P*d*CP). Upon reacting the mixture with gaseous or aqueous NH_3_ in 1941 or primary amine in 1942, P-N and N-P-N were formed (Scheme 2 route **A**, R = Ph). In 1945, P-Ns were made by Atherton, Openshaw, and Todd through a salt elimination process using disubstituted H-phosphonates (H-P), amines, and chlorinating agents (Scheme 2 route **B**) [78]. Subsequently, the Atherton-Todd group synthesized P-Ns via the same method using different polyhalogen solvents in 1947 [79] and using alternative phosphorylating agents in 1948 [80]. In 1951, Wagner-Jauregg et al. synthesized P-N compounds from diisopropylchlorophosphate (*d*PrCP) or tetraethylpyrophosphate (*t*EPyP) and amines (Scheme 2 route **A**, R = *i*-Pr and route **C**, respectively) [81]. Sodium anilide or sodium diphenylamide was refluxed with triphenylphosphate (*t*PPO) to eliminate phenol and give diphenyl *N*-phenylphosphoramidate or diphenyl *N,N*-diphenylphosphoramidate in 1964 (Scheme 2 route **D**) [82]. In 1965, Sundberg reacted 1-ethyl-2-nitrobenzene and triethyl phosphite to obtain a phosphorimidate-product (N=P) that was converted to P-N on silica or alumina columns [83]. Cadogan and coworkers reportedly synthesized and investigated the nucleophilicity of various P-Ns in 1967 [84]. Two years later, a mixture of *N*-arylphosphoramidates, dialkyl phosphoramidates, and phosphonates were obtained when nitroso- and nitro-compounds were reduced in the presence of (RO)_3_P (R = alkyl) [85]. In 1975, Appel and Einig [86] reported a new synthesis to P-Ns by reacting phosphoric acid and an amine in the presence of triphenylphosphine (*t*PP) and CCl_4_. The same year, phase-transfer catalysis was introduced by Zwierzak to synthesize P-Ns through the phosphorylation of amines [87]. The selective alkylation of (EtO)_2_P(O)NH_2_ via silylation was achieved by Zwierzak in 1982 using [*n*-Bu_4_N]Br catalyst (Scheme 2 route **E**) [88]. In 1984, Zwierzak and Osowska-Pacewicka reported the synthesis of various P-Ns via inorganic salt elimination [89] and the direct conversion of phosphoric acid [90]. In 1985, Riesel et al. reacted primary amines or NH_3_ with (RO)_3_P (R = alkyl) and CCl_4_ as an effective route to P-Ns [91]. Nielsen and Caruthers reported the first iodine-mediated reaction of phosphite and *n*-butylamine to synthesize P-Ns in 1988 (Scheme 2 route **F**) [92]. At about the same time, P-Ns were synthesized from a reaction of azide and (RO)_3_P (R = alkyl) in a one-pot process [93]. A magnesium chloride-catalyzed reaction of 2,6-dimethylphenol and phosphorus oxychloride was used to generate phosphorochloridate, which was used for the phosphorylation of amines to synthesize P-Ns [94].

The aforementioned syntheses demonstrate the diversity of organophosphorus synthons available for P-N chemistry up until 1988 and highlight the main pioneers in the field (Figure 4). The research in these initial studies has focused on reducing waste, limiting toxic reagents, improving selectivity, expanding scope, and searching for effective and efficient catalytic P-N bond-making solutions.

### 2.2. Salt Elimination Route

#### 2.2.1. Modifications to the Atherton–Todd Reaction

A synthetic P-N was fortuitously prepared by Atherton, Openshaw, and Todd in 1945 by the salt elimination method using H-phosphonates (H-P) ((RO)_2_P(O)H; R = Et, *i*-Pr, Bn) and ammonia in the presence of a base (Scheme 2 route **B**) [78]. The scope of this method was investigated using different amines (e.g., morpholine, cyclohexylamine, 1-phenylethan-1-amine and benzylamine, 4-methylmorpholine) or a mixture of aniline and *N*,*N*-dimethylcyclohexanamine, as well as different solvents (e.g., CCl_4_, Cl_3_CCCl_3_, and HCl_2_CCCl_3_). Isolated yields of 62–92% were obtained. The Atherton–Todd method was prominent for its *in situ* generation of (RO)_2_P(O)Cl (*d*ACP; R = alkyl, benzyl), which eliminates the use of moisture- and air-sensitive *d*ACP. However, the major drawbacks of the method include the use of halogen sources such as CCl_4_, Cl_3_CCCl_3_, and HCl_2_CCCl_3_ and extensive work-up/purification of the products. In 2020, Chen et al. reported a modified Atherton–Todd reaction for the synthesis of P-Ns using air as a radical initiator (Scheme 3) [95]. The modified method explored P-N synthesis using (H-P) ((RO)_2_P(O)H; R = Et, *i*-Pr, Bu, *i*-Bu) and benzylamine with yields in the range of 30–75%. Although this method offered advantages such as using air as a radical initiator at room temperature, the use of relatively expensive 2,3,4,6,7,8,9,10-octahydropyrimido[1,2-a]azepine (DBU) as a base in the process, the elimination of large amounts of stoichiometric waste, and comparatively lower yields of products are some of the limitations.

#### 2.2.2. Direct Conversion of Diethyl Hydrogen Phosphate

With reference to the limitations (such as low yield and the formation of organophosphorus anhydrides side products) in the method reported by Appel and Einig [86], Zwierzak and Osowska-Pacewicka [90] proposed an alternative method that uses −OH group activator, hexamethyltriaminodibromophosphorane ((Me_2_N)_3_PBr_2_). In the reported method, (Me_2_N)_3_PBr_2_ was prepared in situ in a one-pot two-step synthesis of P-Ns from (EtO)_2_P(O)OH and amines (Scheme 4) [90]. The synthesis was explored using various amines such as aniline, phenylmethanamine, dibutylamine, butan-1-amine, dipropylamine, and hept-1-en-2-amine. This method was favorable with improved yields of P-N products (59–91%) after workup and eliminated the route(s) for anhydride formation. Nevertheless, the method used potentially harmful bromine in a multi-step tedious process with low atom economy.

#### 2.2.3. Inorganic Salt Elimination

In another salt elimination route, this time using inorganic bases and [*n*-Bu_4_N]Br as a catalyst, Zwierzak and Osowska-Pacewicka synthesized P-Ns from (EtO)_2_P(O)H (diethyl H-P) and various amines (Scheme 5a) [89]. Aromatic amines such as aniline and 4-chloroaniline, aliphatic amines such as cyclohexylamine, *n*-butylamine, and ethanamine, and alkanolamines such as 2-aminoethan-1-ol, 3-aminopropan-1-ol, and bis(2-(-ethoxy)ethyl)amine were explored to demonstrate the broad scope of this synthetic transformation. The P-Ns were isolated in 83–100% yield after recrystallization. In a similar synthesis, using [BnEt_3_N]Cl as the catalyst instead of [*n*-Bu_4_N]Br, Zwierzak accomplished the phosphorylation of amines using dialkyl H-phosphonate (RO)_2_P(O)H (R = Et, Bn, *t*-Bu; Scheme 5b) [87]. A range of amines, such as aniline, benzylamine, cyclohexylamine, ethylamine, and diethylamine were used. The crude P-Ns were recrystallized to afford pure products in 35–93% yield. Due to the difficulty encountered in phosphorylating aromatic amines, as mentioned by Zwierzak [87,89], Lukanov et al. proposed a similar synthetic route using *N*-phenylformamide or 2-chloro-*N*-phenylacetamide as the N-source, which offered less steric hindrance and better N-H acidity (Scheme 5c) [96]. Derivatives of *N*-phenylformamide or 2-chloro-*N*-phenylacetamide (R’R’’NC(O)R’’’) were used containing a wide range of aryl-substituents (e.g., R’’ = Ph, 4-MeOPh, 4-ClPh, 2-MeOPh, 2,3-Cl_2_Ph, 2-MePh, and 2,6-(C_2_H_5_)_2_Ph and 2,6-(CH_3_)_2_Ph). Either column chromatography (on silica) or recrystallization were used to purify the P-Ns to obtain isolated yields in the range of 30–80%.

Inorganic salt elimination methods stimulated an era of catalytic transformations of dialkyl H-P ((RO)_2_P(O)H; R = alkyl) to (RO)_2_P(O)Cl (*d*ACP; R = alkyl) in the presence of readily available inorganic bases, thereby eliminating the use of Et_3_N. However, this method has limitations including a pre-functionalization step, the production of large amounts of stoichiometric waste, and the use of hazardous halogenating reagents.

### 2.3. Oxidative Cross-Coupling Route

#### 2.3.1. Using Chlorinating Agents

A one-pot synthesis of P-Ns was reported by Gupta et al. (Scheme 6a) [97]. According to the authors, the chlorination of (RO)_2_P(O)H (R = Me, Et, Pr, *i*-Pr; 3 eq.) to afford (RO)_2_P(O)Cl (*d*ACP; R = alkyl) was possible upon reacting with trichloroisocyanuric acid (*t*CiC-A; 1 eq.), which was then treated with an equal mixture of amines, R’R’’NH (R’ = Me, Et, Pr; R’’ = H) and Et_3_N, to form the P-N in 82–92% yield, which was determined as conversions by ^31^P NMR spectroscopy. However, the isolation of (RO)_2_P(O)NR’R” (R = Et, R’, R” = Et, Pr) product gave 79% yield. A similar method, under base-free reaction conditions, was achieved by Kaboudin et al. (Scheme 6b) [98]. The scope of the synthesis was explored with (H-P) ((RO)_2_P(O)H; R = Et, *i*-Pr) and amine (e.g., *N*-methylaniline, naphthalen-1-amine, 2-phenylethan-1-amine, 3-chloro-2-methylaniline, 3-nitroaniline, benzylamine, or aniline). The crude product was purified by column chromatography (on silica) to yield isolated pure products in the range of 22–92%.

Jang and coworkers reported a facile synthetic route to form P-Ns using diphenyl phosphoric acid (*d*PPA) and an amine in the presence of both a chlorinating agent (Cl_3_CCN) and a base (Et_3_N), which was added to sequester the acid formed during the reaction (Scheme 6c) [99]. The applicability of the method was explored using prop-2-en-1-amine, diethylamine, 2-methylpropan-2-amine, cyclohexylamine, piperidine, morpholine, and aniline. The crude product was purified by column chromatography (on silica) to obtain isolated yields of P-N products in the range of 53–93%. A similar chlorination method reacting an amine (e.g., aniline, hexylamine, diethylamine, and methanamine) with phosphoric acid ((R_2_P(O)OH; R = OEt, OBu, OBn) in the presence of triphenylphosphine and CCl_4_ was also used to synthesize P-Ns (Scheme 6d) [86]. The isolated yields of the products in this case were in the range of 30–75%.

P-N synthesis via oxidative cross-coupling using chlorinating agents offers a short reaction time, low temperature synthesis, and a diverse choice of phosphorylating reagents. Nevertheless, the methods use hazardous chlorinating agents such as CCl_4_ and Cl_3_CCN, generate large stoichiometric wastes, and involve a pre-functionalization step in the synthetic process.

#### 2.3.2. Using Iodinating Agents

The synthesis of P-Ns from amines (aromatic and aliphatic) and symmetrical trialkyl phosphite (P(OR)_3_; R= Me, Et, *i*-Pr) in the presence I_2_ was reported by Chen et al. [100]. Crude products were purified by column chromatography (on silica) and resulted in yields in the range of 0–95%. For the low yielding derivatives, the reaction was more selective to three identified by-products, H-phosphonate, trialkyl phosphate, and tetraalkyl diphosphate, as well as some other unidentified by-products. Important compounds such as phosphoryl amino acid esters (**8**–**11**) were synthesized using the iodine-mediated method (Figure 5).

Iodine (I_2_) has also been used as a catalyst under solvent-free conditions to couple aryl amines (PhNH_2_ and derivatives containing OH, OEt, CN, OMe, NO_2_, F, Cl, or Br substituents on Ph) with (RO)_2_P(O)H (dialkyl H-P; R = Me, Et) affording dialkyl P-Ns (Scheme 7b) [101]. Then, the crude products were purified by column chromatography (on silica) to obtain yields of product in 0–88%. The best yielding conditions were shorter reaction times without solvent. Reactions performed in polar non-protic solvents (e.g., CH_2_Cl_2_), non-polar solvents (e.g., hexane), or protic solvents (e.g., MeOH) all suffered from reduced efficiency and selectivity. Furthermore, aniline with substituents at *ortho*- and *meta*-positions were observed to have reacted faster and were higher yielding than aniline containing substituents at the *para*-position.

I_2_ was used in the presence of H_2_O_2_ as a catalyst at 20 °C for the dehydrogenative cross-coupling of diethyl H-P and amine or sulfoximines to form P-Ns (Scheme 7c) [102]. Purification of the products by column chromatography (on silica) afforded P-Ns with isolated yields of 31–96%, with the lower yielding products reported from sulfoximine. Both alkyl and aryl H-P ((RO)_2_P(O)H, R = Et, *i*-Pr, Ph) were used as the substrate along with cyclic aromatic secondary amines (e.g., indole), derivatives of heterocyclic amines (e.g., furfurylamine), aliphatic cyclic secondary amines (e.g., 1-benzhydrylpiperazine), tertiary amines (e.g., *N*-methylhomopiperazine), or sulfoximine. According to the article, using other sources of I_2_ such as KI or *N*-bromosuccinimide (NBS) for the reaction resulted in decreased yield and trace quantities of the desired products, respectively.

Iodine-mediated oxidative cross-coupling methods offer better routes to P-N compounds by eliminating toxic halogenating agents such as CCl_4_ in the synthetic processes. The method also uses inexpensive oxidants without additives and generates water or alkanol as stoichiometric waste. However, the synthetic routes are prone to the formation of many undesired side products, suffer from low yields, and sometimes have limited applicability.

#### 2.3.3. Using Transition Metal Catalysts

The aerobic oxidative coupling of amines and disubstituted H-phosphonates (H-P) (RO)_2_P(O)H (R = Me, Et, *i*-Pr; 1 eq.) in the presence of copper catalysts to synthesize P-Ns was reported by Hayes and coworkers (Table 1, route a) [103]. The authors reported that the ligands on the copper catalyst were important in the reaction, but not the initial oxidation state of copper catalyst, and that the primary amines coupled better than the secondary and tertiary amines due to the reduced steric hindrance. The method was slightly modified for the synthesis of some important phosphoryl amino compounds such as ethyl (diethoxyphosphoryl)glycinate, etc. (**12**–**15**) (Figure 6).

Another copper-catalyzed aerobic oxidative cross-coupling reaction reported by Wang et al. in 2013 involved the coupling of aryl amines (1.5 eq.) and dialkyl H-phosphonates (RO)_2_P(O)H (R = Et, *i*-Pr, Bu; 1 eq.) in the presence of CuBr (5 mol%) to give P-Ns (Table 1, route b) [104]. Although *para-* or *meta*-substituted aryl amines were very effective substrates in this reaction, the conversions observed with *ortho*-substituted aryl amines or aryl amines with strong electron-withdrawing substituents (e.g., 4-nitroaniline) as well as alkylamines were very poor. The effectiveness of the reaction with diaryl H-phosphonates was not reported.

In 2015, Fe_3_O_4_@MgO nanoparticles were used as a catalyst to synthesize P-N from primary amines and H-P via the Atherton–Todd coupling reaction with CCl_4_ (Table 1, route c) [105]. After column chromatography (on silica), P-Ns were isolated in 52–85% yield. The Fe_3_O_4_@MgO nanoparticles were recycled by washing successively with H_2_O and MeOH and drying before reuse. Although the scope of the transformation was not challenged by investigating aromatic H-Ps, this route offers some advantages, such as the use of a low-cost reusable catalyst with proven activity for up to four consecutive reactions.

Purohit et al. reported a one-pot synthesis of P-Ns from (RO)_2_P(O)H (R = Et, Pr, *i*-Pr; 1 eq.) and amine in the presence of CuCl_2_ and Cs_2_CO_3_ (Table 1, route d) [106]. The reported yields for the variety of desired products were in the range of 25–93% as determined by ^31^P NMR spectroscopy.

A Cu(II) catalyst, a base, and air as an oxidant were used to synthesize *N*-acylphosphoramidates (AcP-N) from (RO)_2_P(O)H (R = Me, Et, *i*-Pr, Bu) and an amide via an oxidative cross-coupling reaction according to Mizuno et al. (Table 1, route e) [107]. The scope of the reaction was investigated by changing the R substituent as well as the nitrogen nucleophiles (e.g., urea, oxazolidinone, indole, pyrrolidinone, lactam, and sulfonamide derivatives). Column chromatography (on silica) using different solvent mixtures was used to purify the desired P-Ns in 52–99% yield. Through the reagent screening, Mizuno et al. [107] found that Cu(OTf)_2_ gave similar cross-coupling results as Cu(OAc)_2_, while other Cu catalysts such as CuCl_2_ and CuSO_4_ and other transition metal catalysts (e.g., Co(OAc)_2_.4H_2_O, Fe(OAc)_2_) were not selective or completely ineffective.

Another copper-catalyzed synthesis of P-N at 20 °C in the presence of air, reported by Zhou et al., involved reacting (RO)_2_P(O)H (R = Me, Et, *i*-Pr, Ph) with primary or secondary amines (butylamine, dibutylamine, hexylamine, cylohexylamine, and methyl alaninate; Table 1, route f) [108]. The crude P-Ns were extracted and then concentrated under vacuum to obtain pure products with excellent isolated yields (86–96%). Similar to other copper-catalyzed P-N synthesis, the authors reported a low yield (21%) when dibutylamine and diisopropyl H-P were used as substrates, which could be attributed to steric effects. In addition, the method was not effective when diphenylphosphine oxide (*d*PPO) was reacted with butylamine (13% isolated yield) due to oxidative conversion to phosphonic acid (**PP-A**). Using this aerobic copper-catalyzed dehydrocoupling method, Zhou et al. synthesized a phosphoryl amino compound (**16**) (Figure 7).

Phenylboronic acid or ester-based substrates were converted *in situ* to an organic azide using a Cu-based catalyst and subsequently used for the synthesis of P-Ns via a coupling reaction, according to Dangroo et al. (Table 1, route g) [109]. Triethyl and trimethyl phosphate were used as the P-sources while phenylboronic acids or esters with substituents such as CN, NO_2_, OH, OMe and halides were used for azide conversion. The reaction was very tolerant to compounds with unsaturated boron–carbon bonds giving isolated yields of 67–93%. However, this reaction is not effective for compounds containing saturated boron–carbon bonds such as butyl boronic acid or cyclopentyl boronic acid.

Transition metal-catalyzed oxidative cross-coupling methods for P-N synthesis have become attractive in recent years because of the advantages of using readily available inexpensive catalysts, for example CuX*_n_* (X = Cl, Br, (OAc), etc., *n* = 1, 2), and air as an oxidant. In addition, this route uses environmentally friendly H-P (RO)_2_P(O)H (R = alkyl, Ph) and amines as starting materials in a one-pot single-step process without any need for purification of the starting material, thereby making the process cost- and time-effective. Apart from eliminating hazardous starting materials, the method uses fewer chemical additives and generates mainly water as a stoichiometric waste. Furthermore, the methods offer a possibility of catalyst recycling, which is an important attribute for any green chemical process in terms of mass production. However, a notable limitation of the synthetic route is the formation of various undesired by-products from the reactivities of H-P and amine with the catalysts under oxidative conditions [103].

#### 2.3.4. Using an Organophotocatalyst

Guo et al. used visible light and organic dyes as catalysts for the dehydrogenative cross-coupling reaction of (EtO)_2_P(O)H (H-P) and amines (aniline, aniline derivatives with CN, X, Me, OMe and 1-naphthylamine) to form the corresponding P-Ns (Scheme 8) [110]. After an extraction, the pure product was formed with isolated yield in the range of 0–99%. It is noteworthy that the method was not effective when vinylaniline and 2-aminothiazole were used because of their radical sensitivity making them prone to either polymerization or decomposition in the reaction media, respectively. Although no product was formed when secondary aliphatic amines were used, moderate yields of up to 59% were obtained from primary or tertiary aliphatic amines, while a complex mixture of products was formed when an aromatic H-P was used.

#### 2.3.5. Using Alkali–Metal Catalyst

An oxidative cross-coupling reaction of amines and (EtO)_2_P(O)H (H-P) in the presence of lithium iodide-tert-butyl hydroperoxide (LiI/TBHP) was used for the synthesis of P-N and other compounds by Reddy et al. (Scheme 9) [111]. The crude products were purified by column chromatography (on silica) with isolated yields of 65% and 31% for the use of benzylamine and 4-chloroaniline, respectively. Other amines were not investigated. Although the method uses an inexpensive catalyst and oxidant, the applicability of the process was not explored using diverse H-P and amines.

### 2.4. Azide Route

Many conventional routes to P-N synthesis as discussed above have used H-P (RO)_2_P(O)H (R = alkyl, Bn, Ph) or (RO)_2_P(O)Cl (*d*ACP; R = alkyl, Bn, Ph) and amine as the P- and N-sources, respectively. As a result, derivatives of P-N compounds synthesized through these routes are limited mainly to substituents on the amine starting materials. Due to increased applicability, it has become necessary to synthesize P-N compounds with diverse functionalities. In order to achieve this, nitrene insertion from organic azides is becoming a popular one because the route uses phosphoryl azide, which acts a dual source of P and N. In this case, it is possible to add (or insert) diverse molecules on the P-N moiety through C-N bond formation.

#### 2.4.1. Nitrene Insertion from Organic Azide

A route to P-N formation through C-N bond formation was explored by Chang, Kim, and coworkers (Scheme 10a) [112]. The P-N was synthesized in the presence of an Ir(III)-based catalyst using an amide or a ketone and a phosphoryl azide. The product was purified by column chromatography (on silica) with isolated yields in the range 41–99%. The scope of the synthesis was explored using phosphoryl azide substituted with binaphthyl, aryl, and alkyl groups, while the phenone had substituents such as phenyl, methyl, and isobutane groups. Furthermore, 4-methoxy, 4-CF_3_, and tert-butyl were some of the substituents on the benzamide. Biologically active phosphoramidates (**17** and **18**) were synthesized through this method (Figure 8).

Che and coworkers later introduced a Ru^IV^-porphyrin-catalyst to synthesize *N*-acylphosphoramidates (AcP-N) from aldehydes and phosphoryl azide through nitrene insertion (Scheme 10b) [113]. The product was isolated by column chromatography (on silica). The observed product conversions were in the range 56–99%. Aromatic and aliphatic aldehydes such as 2-naphthaldehyde, benzo[d][1,3]dioxole-5-carbaldehyde, 5,6-dihydro-2H-pyran-3-carbaldehyde and 3-methylbut-2-enal were used in the reaction along with aromatic and aliphatic phosphoryl azide such as 3-nitrophenyl (4-nitrophenyl) phosphorazidate, bis(2,2,2-trichloroethyl) phosphorazidate, and diethyl phosphorazidate. However, lower yields were obtained from aliphatic aldehydes when compared with the aromatic aldehydes, suggesting that the synthetic procedure favors electron-rich aromatic aldehydes.

A nitrene insertion route in the presence of Ir(III)-based catalyst was also used to synthesize P-Ns by Zhu et al. (Scheme 10c) [114]. The P-Ns were synthesized under an inert environment and the isolated yield of the products was in the range of 52–77%. The scope of the synthesis was investigated using diethyl phosphorazidate and diphenyl phosphorazidate along with 2-phenylpyridine or pyrazole derivatives with substituents such as Me, OMe, CF_3_, Cl, F, Ph, and COOMe at *ortho*-, *meta*-, and *para*-positions as well as benzo[h]quinoline.

Although nitrene insertion provides a viable route to P-N synthesis and generates N_2_ gas as the only stoichiometric by-product, the process uses expensive catalysts in addition to many chemical additives. Moreover, apart from the hazards associated with the handling of phosphoryl azide, the route requires a long reaction time, high temperature, and does not favor electron-deficient starting materials.

#### 2.4.2. Via In Situ Azide Generation

An organic azide was generated in situ from organic halides and was coupled with (RO)_3_P (R = alkyl) for the synthesis of P-N by Sangwan et al. (Scheme 11) The crude product was purified by column chromatography (on silica) to afford isolated products in the range of 52–96% yield. A broad scope of the reaction was tested; 3-bromoprop-1-yne, 3-bromoprop-1-ene, 1-bromopropane alkyl or (un)substituted aromatic halide such as (bromomethyl)benzene, 1-(bromomethyl)-2-nitrobenzene, 1-(chloromethyl)-4-nitrobenzene, (chloromethyl)benzene, 1-(bromomethyl)-3,5-dimethoxybenzene, or 3-(4-(bromomethyl)-1H-1,2,3-triazol-1-yl)benzyl acetate were reacted with dimethyl or diethyl H-P.

A similar method of P-N synthesis via in situ azide generation from organic halides was reported by Dar [115], following the procedure previously reported by Sangwan et al. (Scheme 11) [116]. The yield range of the isolated P-Ns, in this case, was 60–96%. Phosphoramidate-derived bioactive compound (**19**) was synthesized using the proposed method (Figure 9).

#### 2.4.3. Via Two-Step Organic Azide Generation

In 1988, Koziara reported a two-step P-N synthetic route via organic azide generation using a [*n*-Bu_4_N]Br catalyst (Scheme 12) [93]. The crude product was purified by crystallization or vacuum distillation with 50–93% isolated yield. The organic halides used were (3-bromopropyl)benzene, (2-bromoethyl)benzene, 1-bromohexane, bromocyclohexane, 1-bromobutane, or bromocyclopentane.

Routes via organic azide generation use readily available, less toxic organic halides and (RO)_3_P (R = alkyl) relative to other routes and generate N_2_ as stoichiometric waste. However, the method relies on the pre-functionalization and preparation of hazardous azide precursors in multi-step processes.

#### 2.4.4. Transition Metal-Free Synthesis from Organic Azide

Transition metal-free synthesis of P-N from phosphoryl azide and amines was accomplished by Chen et al. [117]. The synthetic process uses the nitrogen atom available from the amine rather than from the azide (Scheme 13). The residue was purified by column chromatography (on silica) to afford the isolated product in 52–92% yield. The scope of the reaction was examined by using diphenyl and diisopropyl phosphoryl azide along with primary amines (e.g., isopropylamine and *t*-butylamine), substituted primary amines (e.g., thiophen-2-ylmethylamine and 4-chlorobenzylamine), and heterocyclic amines (e.g., pyrrolidine and morpholine). Although the method circumvents the use of metal catalysts in the synthesis, the use of a hazardous azide and high temperature are its limitations.

### 2.5. Reduction Route

#### 2.5.1. Via Nitro-Group Reduction

Beifuss et al. reported a microwave-assisted synthesis of *N*-arylphosphoramidates from nitrobenzene and (RO)_3_P (R = alkyl) (Scheme 14) [118]. The crude residue product was purified by column chromatography (on silica) resulting in pure P-Ns with 52–79% isolated yield. The scope of the synthesis was investigated using trimethyl or triethyl phosphite along with nitrobenzene, and mono- or di-substituted nitrobenzenes with substitution such as Me, OMe, Cl, Br, or CN groups. This method is attractive because it provides a different route to P-Ns using nitro compounds instead of the conventional amines. Nonetheless, the high temperature and amount of energy required during the synthesis are the obvious limitations.

#### 2.5.2. Via Catalyst-Free Staudinger Reduction and Lewis-Acid Catalyzed Rearrangement

Hackenberger et al. synthesized P-N compounds from organic azide derivatives and (RO)_3_P (R = alkyl) after Lewis acid-catalyzed rearrangement (Scheme 15) [119]. The crude product was concentrated under reduced pressure, and the desired product was isolated (63–98% yield) through column chromatography (on silica). Triethyl or trimethyl phosphite were used as the P-source along with organic azide derivatives such as (azidomethyl)benzene, azidocyclohexane, (2-azidopropan-2-yl)benzene, azidobenzene, ethyl 2-azidoacetate, 1-azidodecane, (azidomethylene)dibenzene, and (3-azidoprop-1-en-1-yl)benzene.

### 2.6. Hydrophosphinylation Route

Zimin et al. reported the 1,2-hydrophosphinylation of alkyl nitriles (R’-CN, R’ = alkyl, Ph, Bn) using sodium dialkyl phosphite ((RO)_2_PONa, R = Et, Pr) to achieve the formation of a P-N featuring a P-C-N-P linkage in which the dialkyl phosphite group of (RO)_2_PONa was added to the carbon and nitrogen atoms of the nitriles (Scheme 16) [120].

### 2.7. Phosphoramidate-Aldehyde-Dienophile (PAD) Route

Chabour et al. reported a diastereoselective P-N synthesis via the PAD process using a TsOH acid catalyst and a dienophile, maleimide (Scheme 17) [1]. Flash chromatography was used to isolate the desired product in 40–88% yield. The scope of the synthesis was explored with maleimide derivatives with substituents such as 4-FC_6_H_4_CH_2_, C_6_H_4_CH_2_, C_6_H_4_COCH_3_, 4-BrC_6_H_4_, phenyl, and methyl. The unsaturated aldehyde derivatives that were used were 3,7-dimethylocta-2,6-dienal, 3-methylbut-2-enal, pent-2-enal, and hex-2-enal. This newest route for P-N synthesis uses inexpensive catalysts, generates only H_2_O as a by-product in stoichiometric amounts, and demonstrates an important route to the synthesis of polyfunctionalized P-N compounds.

## 3. Mechanistic Considerations

A wealth of information on the mechanisms for the synthetic routes to P-Ns is available. For each of the main categories highlighted above, researchers have provided mechanistic insight into the reaction by determining the main sources of oxidation as well as electronic and steric factors provided from the substituents that influence the reaction rate. Several examples of postulated catalytic cycles will be provided below, illustrating the key steps, similarities, and differences between them.

### 3.1. Salt Elimination Route

#### 3.1.1. Atherton–Todd Reaction

Atherton, Openshaw and Todd suggested that the two-step mechanistic process for the salt elimination could involve the formation of dialkyl or dibenzyl (trichloromethyl)phosphonate in the first step, from a reaction of (RO)_2_P(O)H (H-P; R = alkyl) and CCl_4_ in the presence of Et_3_N [78]. The desired product is formed in the second step when the (trichloromethyl)phosphonate reacts with amine, forming also CHCl_3_ in the process (Scheme 18). Although this mechanism is plausible, evidence by NMR spectroscopy suggests that in the first step, a reaction of the base and CCl_4_ results in salt formation [Et_3_NCl]^+^[CCl_3_]^−^ [121]. The H-P is deprotonated by [CCl_3_]^−^ to form CHCl_3_ and anionic phosphite, which reacts further with [Et_3_NCl]^+^ to form (RO)_2_P(O)Cl (*d*ACP; R = alkyl). In the second step, *d*ACP reacts with amine to give the product.

For the modified Atherton–Todd reaction, the two-step mechanistic process involves the deprotonation of H-P by DBU, and the radical formed is autoxidized by O_2_ before it abstracts chlorine from CHCl_3_ to form *d*ACP (Scheme 19) [95]. The nucleophilic substitution of *d*ACP by the amine affords the P-N product.

#### 3.1.2. Direct Conversion of Diethyl Hydrogen Phosphate

In the plausible mechanism for the direct conversion of diethyl hydrogen phosphate, hexamethyltriaminophosphine P(NMe_2_)_3_ is oxidized to hexamethyltriaminodibromophosphorane ((Me_2_N)_3_PBr_2_) which is ionized in solution (Scheme 20) [90]. Subsequently, the ionized (Me_2_N)_3_PBr_2_ and diethyl hydrogen phosphate undergo ligand exchange in the presence of Et_3_N to form the intermediate, diethyl (phosphaneyloxy)phosphonate amino-hydrazine-bromide salt complex. Nucleophilic attack from the amine onto the positively charged P results in the diethyl phosphoramidate bromide salt, which reacts with Et_3_N to give the desired product with a concomitant elimination of triethylammonium bromide. Apart from the stoichiometric Et_3_N salt waste generated, the synthetic process is prone to the formation of significant quantities of pyrophosphate or phosphanetetraamine bromide salt side products, resulting in a low yield.

The mechanistic steps for the synthesis of P-N via a salt elimination route were not proposed by Zwierzak et al. [89]; however, based on the roles of KHCO_3_, K_2_CO_3_, and [*n*-Bu_4_N]Br as outlined in the paper and a recently proposed mechanism for an Atherton–Todd type reaction, a probable mechanism can be postulated (Scheme 21) [121]. In the reacting system, the HCO_3_^−^ migrates to the organic phase in the presence of [*n*-Bu_4_N]Br as the catalyst, where it is protonated and reacts with CCl_4_ to form anionic HCO_3_ClH^−^ and cationic Cl_3_C^+^. The dialkyl H-phosphonate ((RO)_2_P(O)H, R = alkyl) is deprotonated by the cationic Cl_3_C^+^ to yield CHCl_3_ and anionic phosphate, (RO)_2_P(O)^−^. Next, the anionic phosphate reacts with HCO_3_ClH^−^ to form (RO)_2_P(O)Cl (*d*ACP; R = alkyl) and H_2_CO_3_. In the following step, a nucleophilic substitution between the amine and *d*ACP results in the formation of the desired product. Meanwhile, K_2_CO_3_ serves as the dehydrating agent in the reacting system. The plausible mechanism is also similar when NaOH and [BnEt_3_N]Cl are used in place of KHCO_3_ and [*n*-Bu_4_N]Br, respectively.

Lukanov et al. suggested that the mechanistic cycle for their synthesis involves the deprotonation of dialkyl H-phosphonate ((RO)_2_P(O)H, R = alkyl) to form (RO)_2_P(O)Cl (*d*ACP, R = alkyl) [96]. Then, the *d*ACP reacts with *N*-phenylformamide or the 2-chloro-*N*-phenylacetamide derivative to form the product. Alternatively, the mechanism could proceed via a dialkyl (acetyl)(phenyl)phosphoramidate intermediate, which subsequently undergoes hydrolysis to afford the product.

### 3.2. Oxidative Cross-Coupling Route

#### 3.2.1. Using Chlorinating Agents

The mechanisms of P-N formation from Gupta et al. [97] and Kaboudin et al. [98] suggest that the dialkyl H-phosphonate ((RO)_2_P(O)H, R = alkyl) reacts with trichloroisocyanuric acid (*t*CiC-A) to form the intermediate (RO)_2_P(O)Cl (*d*ACP; R = alkyl) and 1,3,5-triazinane-2,4,6-trione (*t*A*t*O) (Scheme 22). Following, a nucleophilic substitution between the *d*ACP and the amine result in the P-N with HCl as the major by-product.

The mechanism of the reaction by Jang et al. suggests that diphenyl phosphoric acid (*d*PPA) undergoes chlorination in the presence of a base, which is subsequently attacked by the amine to afford the P-N product [99]. Although the authors did not propose any mechanism, they suggested that nucleophilic attack on the P atom was consistent with aliphatic amines resulting in a higher yield of products than the aromatic amines because aliphatic amines are more nucleophilic.

The mechanistic step suggested by Appel and Einig involves the formation of oxo-triphenyl-diphosphoxanium salt intermediate from a deprotonation of phosphoric acid by CCl_4_ in the presence of phosphine (Scheme 23) [86]. Nucleophilic substitution from amine results in the desired product and triphenylphosphate as the by-product.

#### 3.2.2. Using Iodinating Agents

In the suggested reaction mechanism by [100], iodine reacts with (RO)_3_P (R = alkyl) to form trialkoxyiodophosphonium intermediate, which is further attacked by iodide ion to form dialkyl phosphoriodidate (*d*APPI) (Arbuzov reaction, Scheme 24). Then, the *d*APPI is attacked by the amine to give the desired P-N with an elimination of HI. *N*-phosphoryl amino acid esters and P-N with a free amino group were also synthesized using the iodine-mediated synthesis method. The challenges with this iodine-mediated P-N formation is that many side products are formed, and the reaction procedure does not generate the desired product when *N*-iodosuccinimide (NIS) is used as the iodine source. In addition, the reaction process is ineffective with aromatic H-phosphonates because the reaction is affected by steric effects on the substituents.

In a mechanism of the I_2_-mediated P-N formation, according to Singh et al., H-P reacts with I_2_ to form an intermediate, which undergoes nucleophilic attack from the amine to generate a hydroxylated-dialkyl phenylphosphoramidate complex with an elimination of I_2_ (Scheme 25) [101]. The phosphoramidate complex further undergoes dehydrogenation to eliminate H_2_ in form of H_2_O and afford the desired products.

The suggested mechanism of a similar reaction reported by Prabhu et al. instead involves the nucleophilic substitution of phosphoriodate intermediate after the electrophilic iodination of the H-P starting material (Scheme 26) [102]. Then, the iodide ion is oxidized to active catalyst, hypoiodous acid, in the presence of H_2_O_2_.

#### 3.2.3. Using Transition Metal Catalyst

In a copper-catalyzed oxidative cross-coupling reaction, Hayes et al. proposed a reaction mechanism involving single-electron transfer oxidation of the copper catalyst, which favored the formation of the Cu(II)-amine complex (Scheme 27) [103]. The Cu(II)-amine complex is further oxidized to form the Cu(III)-amine-phosphonate complex, which undergoes reductive elimination of the copper catalyst to yield the desired P-N. Although this mechanism seems plausible, it could not easily explain the formation of the side products.

There were no suggested mechanisms for the reaction in the report by Wang et al. [104]. However, the results of the reagent screening for the reaction had suggested that other transition metal catalysts (e.g., CoBr_2_) and other Cu catalysts (e.g., Cu(OAc)_2_·H_2_O) were inactive or inefficient for the catalysis. It was further suggested that although a CuCl catalyst enabled reaction, it was not as effective as CuBr and conducting the reaction under anaerobic conditions (e.g., under N_2_) did not lead to the desired product.

In the Fe_3_O_4_@MgO nanoparticle-mediated synthesis of P-Ns as per the procedure reported by Habibi et al. [105], the mechanism involves the formation of the (RO)_2_P(O)Cl (*d*ACP; R = alkyl) intermediate from the reaction of dialkyl H-phosphonate ((OR)_2_P(O)H, R = alkyl) with CCl_4_ (Scheme 28). Following, the amine in the presence of Fe_3_O_4_@MgO nanoparticles couples with the *d*ACP by nucleophilic substitution to give the P-N product, eliminating HCl.

According the plausible two-step mechanism for the reaction by Purohit et al., (RO)_2_P(O)Cl (*d*ACP; R = alkyl) is first generated from the electrophilic displacement of hydrogen on the dialkyl H-phosphonate ((OR)_2_P(O)H, R = alkyl) upon reacting with CuCl_2_ (Scheme 29a) [106]. The phosphoramidate is formed in the second step by coupling *d*ACP and amine. However, this reported mechanism could not properly explain the role of Cs_2_CO_3_ in the coupling of alkyl chlorophosphate and the amine to give the P-N. This is because the coupling reaction was expected to eliminate other products along with CsCl such as a highly reactive CsH and a carbonate. However, it is to be understood that the use of Cs_2_CO_3_ as the inorganic base was based on the poorly understood “caesium effect” in which the highly soluble caesium base selectively favors the formation of one functional group in preference to another [122,123].

Mizuno et al. proposed a mechanism for the reaction in which K_2_CO_3_ first deprotonates the amide, affording unpaired electrons on a Cu-amide complex [107]. Furthermore, it was suggested that it is possible that K_2_CO_3_ deprotonates H-P or provides a medium for the H-P tautomer to further complex with the Cu-amide complex. The Cu-amide-phosphite intermediate undergoes reductive elimination to afford the desired product, while O_2_ re-oxidizes the reduced Cu species to regenerate the active catalyst. Although the oxidation state of Cu in the reaction mechanism was not explained, the authors suggested that it could be +2, and the fact that the reaction produced only stoichiometric amounts of the desired product (relative to the Cu catalyst) under an inert atmosphere was a support to the re-oxidation of the Cu species in air.

From their experimental observations, Zhou et al. strongly suggested that the oxidative dehydrocoupling reaction for P-N synthesis using Cu-catalysts proceeds stereospecifically through an inversion of the configuration at P [108]. This stereospecific inversion chemistry was used as an argument in the proposed mechanism to support a formation of halogen–phosphate intermediate when Cu catalyst is reacted with H-phosphonate (H-P). The amine (as a nucleophile) subsequently attacks the intermediate from behind to give the product, while the Cu(I) catalyst is then reoxidized by O_2_ to Cu(II) in the presence of hydrogen halide formed from the process. Water is also formed as a stoichiometric waste (Scheme 29b).

The mechanism as reported by Dangroo et al. involved transmetalation to form an azidocopper(II) species in the first step (Scheme 30) [109]. Similar transmetalation reactions involving the phenylboronic acid-based substrates and the azidocopper(II) species in the second step result in the formation of a phenyl–azidocopper(II) complex. The complex is oxidized in the presence of Cu(II) to a phenyl–azidocopper(III) complex, which results in the formation of the organic azide and forms Cu(I) after reductive elimination. From this, a reaction between the (RO)_3_P (R = alkyl) and the organic azide gives phosphorimidate (with loss of N_2_), which undergoes hydrolysis to afford the desired product, while the Cu(I) is reoxidized to Cu(II) in the catalytic cycle. Although the mechanism satisfies much of the observed results, the proposed second step, transmetalation, would have resulted in borazidic acid (i.e., N_3_-B(OH)_2_) and Cu(II)-phenyl species or Cu(OH)_2_ species when the azidocopper(II) species reacted with phenylboronic acid-based substrates, based on the exchange of ligands as in the first step.

#### 3.2.4. Using an Organophotocatalyst

In a suggested mechanism for the organophotocatalytic synthesis of P-N, a photon from the visible light source is absorbed by the organic dye to become an excited radical (Scheme 31) [110]. By a single-electron transfer process, the excited dye-radical abstracts an electron from the N atom of the amine to form an anionic dye radical and cationic amine radical. In the presence of O_2_, the anionic dye radical is reoxidized to the organic dye, while the anionic superoxide that is formed in the process reacts with diethyl H-phosphonate (H-P) to generate anionic hydrogen peroxide and a diethyl phosphoryl radical. Then, the radical reacts with the cationic-amine radical to form a protonated P-N, which is then deprotonated by the anionic hydrogen peroxide. The authors reported that after an initial 74% isolated product, the organic dye catalyst was recycled a second time using an acid–base extraction, and 67% isolated product was obtained the second time.

#### 3.2.5. Using Alkali–Metal Catalyst

In the overall mechanism suggested for the alkali–metal catalyzed oxidative cross-coupling reaction, a phosphoryl iodide intermediate and LiOH are generated in the first step from a reaction of LiI and the H-phosphonate (H-P) in the presence of aq. TBHP (Scheme 32) [111]. The second step involves the removal of a proton from the amine by the LiOH to form lithium amide and H_2_O, and the desired product is formed in the final step from the reaction of phosphoryl iodide and lithium amide.

### 3.3. Azide Route

#### 3.3.1. Nitrene Insertion from Organic Azide

The mechanism of the reaction of nitrene insertion according to Chang, Kim, and coworkers follows activation of the C-H bond cleavage on the amide or ketone substrate by the Ir^(III)^-based catalyst to generate an iridacyclic intermediate, which then reacts with the phosphoryl azide to produce a coordinated phosphoryl azide–iridacyclic adduct (Scheme 33) [112]. By intramolecular migratory insertion of C-N into the Ir-C bond, with removal of N_2_, or by nucleophilic attack from the azide group on the Ir-C bond to form the C-N bond with the elimination of N_2_, a complex of Ir-N-C bond is formed. Proto-demetallation of the Ir^(III)^-based catalyst affords the desired product while regenerating the catalyst.

According to Zhu et al., the mechanistic steps involve activation of the Ir^(III)^-based catalyst in the presence of AgOAc and AgSbF_6_, which abstracts an H atom from the 2-phenylpyridine derivative to cyclo-(2-phenylpyridine)iridium(III) complex [114]. The complex further coordinates with phosphoryl azide to form a cyclo-(2-phenylpyridine)iridium(III)-phosphoryl azide complex, which undergoes migratory insertion to give a complex containing a C-N-Ir^(III)^ bond, releasing N_2_. The Ir^(III)^-based catalyst is released through proto-demetallation to generate the desired product.

For the nitrene insertion using Ru^IV^-porphyrin-based catalyst, Che and coworkers proposed that a reactive intermediate Ru^IV^-porphyrin-phosphorylimide species is first formed from the interaction of phosphoryl azide and the Ru^IV^–porphyrin catalyst [113]. Then, the intermediate reacts with the aldehyde C-H bond through nitrene insertion to form a C-N bonded Ru^IV^-porphyrin-phosphorylimide-aldehyde complex with the removal of N_2_. The desired *N*-acylphosphoramidates (AcP-N) product is formed after proto-demetallation of the Ru^IV^-porphyrin catalyst, and the catalyst is regenerated in the process.

#### 3.3.2. Organic Azide Generation Route

In the reported mechanisms of the reaction for P-N synthesis via azide generation, organic azide is formed in the first step from nucleophilic substitution of the organic halide with the elimination of N_2_ gas. In the second step, the organic azide reacts with (RO)_3_P (R = alkyl) to form the P-N compound after rearrangement or hydrolysis (Scheme 34) [93,115,116].

According to the mechanism suggested by Chen et al., the amine reacts with phosphoryl azide through a nucleophilic substitution reaction to generate an azido (hydroxy)-phosphanamine complex intermediate in the first step [117]. In the second step, the azide anion departs as the leaving group to give a stable P-N product (Scheme 35). Although the mechanism looks highly plausible, it is noteworthy that N_3_ is highly nucleophilic because in its canonical structure, one of the nitrogen atoms carries a double negative charge (N≡N^+^ − N^2−^) [124], which makes it stable and a poor leaving group as N_3_ in the presence of amine [125]. In addition, the mechanism does account for the potentially explosive HN_3_ in a closed flask at 120 °C. However, it is possible that in the reaction process, the azide is first reduced or decomposed to a nitrene phosphonate radical ((RO)_2_P(O)N; R = alkyl, Ph), while N_2_ is liberated. In the second step, the electrophilic nitrene phosphonate radical reacts with or is substituted by the nucleophile amine to give the product.

### 3.4. Reduction Route

#### 3.4.1. Via Nitro-Group Reduction

The proposed mechanism of the reaction for the nitro-group reduction route involves the reduction of nitrobenzene in the presence of (RO)_3_P (R = alkyl) to give nitrosobenzene with a phosphate side product (Scheme 36) [118]. In the second step, the nitrosobenzene is reduced further in the presence of another equivalent of (RO)_3_P, generating phenylnitride with another phosphate side product. A further reaction of (RO)_3_P (R = alkyl) with the basic phenylnitride generates a trialkyl phenylphosphorimidate intermediate, which undergoes hydrolysis to give the desired *N*-arylphosphoramidate product. Experiments to verify the proposed mechanism using aniline in the presence of excess triethyl phosphite or triethyl phosphate did not yield the desired product under the optimized conditions, suggesting that the reaction proceeds via reduction of the nitro group to N^2−^ and that the product is not formed from phosphate.

#### 3.4.2. Via Catalyst-Free Staudinger Reduction and Lewis-Acid Catalyzed Rearrangement

The plausible two-step mechanistic process suggested by Hackenberger’s group for the synthesis involves the formation of phosphorimidate from the reduction of azide in the presence of (RO)_3_P (R = alkyl), eliminating N_2_, in the first step (Scheme 37) [119]. In the second step, BF_3_ activates the nitrogen site of the phosphorimidate by forming a coordinated complex. Electrophilic attack by a neighbouring alkyl group of phosphorimidate on the negatively charged N atom of the phosphorimidate-BF_3_ complex results in an ((alkyl)amino)trialkoxyphosphonium complex, cleaving off BF_3_. Nucleophilic attack on the trialkoxyphosphonium complex from a neighboring phosphorimidate completes the catalytic cycle, affording a P=O bond of the desired P-N product.

### 3.5. Hydrophosphinylation Route

For the hydrophosphinylation route to P-Ns, Zimin et al. proposed that in the first step, sodium dialkyl phosphite ((RO)_2_PONa, R = Et, Ph) reacts with alkyl nitrile to form an imine (C=N), which reacts further with a second equivalent of (RO)_2_PONa to form the 1,1-addition product, sodium (1,1-bis(dialkoxyphosphoryl)alkyl)amide (Scheme 38) [120]. In the second step, sodium (1,1-bis(dialkyoxyphosphoryl)alkyl)amide undergoes hydrolysis and isomerization to give the 1,2-addition phosphoramidate product containing a P-C-N-P bond, eliminating a sodium cation in the process.

### 3.6. Phosphoramidate-Aldehyde-Dienophile (PAD) Route

Chabour et al. proposed a mechanism of the reaction in which (EtO)_2_P(O)NH_2_ reacts with (or is activated by) acetic anhydride to form diethyl acetylphosphoramidate and acetic acid (Scheme 39) [1]. Meanwhile, the aldehyde protonated in the presence of an acid catalyst, TsOH, reacts with the diethyl acetylphosphoramidate complex to give the cationic diethyl (alkene-acetamido)phosphonate complex intermediate. The unstable cationic complex undergoes rearrangement with a hydride shift and deprotonation in presence of TsO^−^ to give a more stable diethyl acetyl(alk-1,3-diene)phosphoramidate. The subsequent direct addition of the maleimide derivative results in a P-N adduct intermediate, which is deacetylated in the presence of acetic acid to afford the desired product.

## 4. Summary

In this review, the applications, syntheses, and corresponding mechanisms of phosphoramidates have been discussed in depth. As demonstrated from the many categories for the uses of synthetic P-Ns, it is clear that the P-N bond motif has structural importance in diverse fields. Early synthetic routes to P-Ns are limited due to harsh reaction conditions, air and moisture-sensitive reagents, multi-step synthesis, and undesirable side reactions. The separation and purification of P-Ns is also a challenge. In particular, most of the synthetic strategies involve extraction and/or column chromatography, which not only requires large amounts of solvent, but also takes a significant amount of time. Therefore, unsurprisingly, extensive studies have been undertaken (with more to come in the future) for more sustainable, efficient, atom economic, selective, and milder synthetic procedures toward P-Ns. In addition, the discussed mechanisms require a deeper understanding of the associated phosphorus chemistry in order to have better synthetic control and to optimize the yield and selectivity toward P-N products. To overcome these challenges, it is predicted that better catalysts, improved reaction conditions, and new solvent solutions for separating the product from the by-products and potentially improving selectivity are in the forecast. For example, there is a sharp rise into synthetic research under solvent-free conditions or using ionic liquids (ILs) and deep eutectic solvents (DES). These can be environmentally benign, highly versatile, selective, and recoverable solvents for use in catalysis.

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
