# Peer review of "Opening up the Toolbox: Synthesis and Mechanisms of Phosphoramidates"

_molecules, 2020, doi:10.3390/molecules25163684_

Round 1

Reviewer 1 Report

The review entitled "Opening Up the Toolbox: Synthesis and Mechanisms 2 of Phosphoramidates" compiles data on preparation of organophosphorus compounds.

There are some little mistakes 

Page 2:

- Lines 40 to 61 The structures should be numbered in separated parenthesis (1-7) (Fig. 1.2)

- Lines 55 to 57 Pesticides is a generic term which includes insecticides, fungicides, herbicides and parasiticides, therefore this part in the text needs a little revision.

Figure 1.3  the role of the molecules at the agriculture and MALDI-TOF sections needs to be mentioned to made the figure more uniform

Page 4

Line 91 " molecular weight such as amino acids and peptides,..."

Page 5

line 147 "A Brief Historical Timeline of Phosphoramidate Synthesis" lack of the letter r

References 20 and 31 lack DOI

The authors should provide a critical and comparative discussion on the six synthetic routes presented. Which methods give good yield and sound efficient, less toxic, most ecofriendly and cheaper. 

Some hit examples of synthesis can also be shown  

Author Response

Many thanks for your very helpful comments.  The little mistakes on pages 2, 4, 5 and References 20/31 have been changed in the manuscript.  We have revised section 2 to accommodate your useful suggestion regarding a critical discussion of the advantages and drawbacks of the various methods. Figures 2.1-2.5 have been added with hit examples of P-Ns.

Reviewer 2 Report

The manuscript is a review devoted to the synthesis and mechanistic discussions of phosphoramidates. As such, the manuscript provides a good insight into the chemistry of this class of compounds and could be accepted for publication in Molecules journal. The only concern here is too detailed description of the synthetic part where the detailed synthetic procedure is cited. The review itself must be rather a critical discussion of the available data, so my suggestion to the Auhors i to re-edit the first part of the manuscript devoted to the synthesis to provide more general discussion about the synthetic pathways.

Author Response

Many thanks for your valuable comments.  We have substantially re-written section 2 to provide more general comments on the synthetic pathways to P-Ns.

Round 2

Reviewer 2 Report

The manuscript has been improved in a new version and should be accepted for publication in Molecules journal.